# A Transposon Story: From TE Content to TE Dynamic Invasion of *Drosophila* Genomes Using the Single-Molecule Sequencing Technology from Oxford Nanopore

**DOI:** 10.3390/cells9081776

**Published:** 2020-07-25

**Authors:** Mourdas Mohamed, Nguyet Thi-Minh Dang, Yuki Ogyama, Nelly Burlet, Bruno Mugat, Matthieu Boulesteix, Vincent Mérel, Philippe Veber, Judit Salces-Ortiz, Dany Severac, Alain Pélisson, Cristina Vieira, François Sabot, Marie Fablet, Séverine Chambeyron

**Affiliations:** 1Institute of Human Genetics, UMR9002, CNRS and Montpellier University, 34396 Montpellier, France; mourdas.mohamed@igh.cnrs.fr (M.M.); yuki.ogiyama@igh.cnrs.fr (Y.O.); bruno.mugat@igh.cnrs.fr (B.M.); alain.pelisson@igh.cnrs.fr (A.P.); 2IRD/UM UMR DIADE, 911 avenue Agropolis BP64501, 34394 Montpellier, France; dangminhnguyet09@gmail.com (N.T.-M.D.); francois.sabot@ird.fr (F.S.); 3Université de Lyon, Université Lyon 1, CNRS, Laboratoire de Biométrie et Biologie Evolutive UMR 5558, 69622 Villeurbanne, France; nelly.burlet@univ-lyon1.fr (N.B.); matthieu.boulesteix@univ-lyon1.fr (M.B.); vincent.merel@etu.univ-lyon1.fr (V.M.); philippe.veber@univ-lyon1.fr (P.V.); judit.salces@ibe.upf-csic.es (J.S.-O.); cristina.heddi@univ-lyon1.fr (C.V.); 4Institute of Evolutionary Biology (IBE), CSIC-Universitat Pompeu Fabra, 08003 Barcelona, Spain; 5MGX-Montpellier GenomiX, c/o Institut de Génomique Fonctionnelle, CNRS, INSERM, Université de Montpellier, 34094 Montpellier, France; Dany.Severac@mgx.cnrs.fr

**Keywords:** transposable elements, ONT, *Drosophila melanogaster*, *Drosophila simulans*, piRNA

## Abstract

Transposable elements (TEs) are the main components of genomes. However, due to their repetitive nature, they are very difficult to study using data obtained with short-read sequencing technologies. Here, we describe an efficient pipeline to accurately recover TE insertion (TEI) sites and sequences from long reads obtained by Oxford Nanopore Technology (ONT) sequencing. With this pipeline, we could precisely describe the landscapes of the most recent TEIs in wild-type strains of *Drosophila melanogaster* and *Drosophila simulans*. Their comparison suggests that this subset of TE sequences is more similar than previously thought in these two species. The chromosome assemblies obtained using this pipeline also allowed recovering piRNA cluster sequences, which was impossible using short-read sequencing. Finally, we used our pipeline to analyze ONT sequencing data from a *D. melanogaster* unstable line in which LTR transposition was derepressed for 73 successive generations. We could rely on single reads to identify new insertions with intact target site duplications. Moreover, the detailed analysis of TEIs in the wild-type strains and the unstable line did not support the trap model claiming that piRNA clusters are hotspots of TE insertions.

## 1. Introduction

Transposable elements (TEs) are major components of almost all eukaryotic genomes [1,2]. They can be separated into three main groups that include several TE superfamilies and families: DNA transposons, Long-Terminal Repeat (LTR) elements, and Long Interspersed Nuclear Elements (LINEs) [2,3]. Different methods (e.g., Southern blotting [4,5], in-situ hybridization on polytene chromosomes [6,7], and PCR [8,9]) were first used to estimate TE content in *Drosophila* genomes and to understand how TEs invade and shape genomes by affecting genome function and evolution. However, technical problems linked to TE repetitive nature and diversity have not allowed for the reaching of firm conclusions and many questions about TE biology remain unanswered.

Then, next-generation short-read sequencing technologies allowed for characterizing the global TE content within and between related species. Moreover, the high coverage provided by Illumina sequencing led to the identification of consensus sequences for each TE family. Several computational methods were developed, such as RepeatExplorer [10] and dnaPipeTE [11], to analyze Illumina data from different *Drosophila* species, and to study TE biology at the populational level.

In TE biology, it is also important to estimate the TE insertion (TEI) rate to determine the degree of polymorphism within and between populations. This is an indicator of the activity level of each TE family and can help to date transposition events [12,13]. Illumina sequencing of pools of individuals allowed for determining TEI frequency in natural samples (from large numbers of individuals to populations) [14]. To study individual TEI, dedicated software tools were developed (e.g., TIDAL [15], T-Lex/T-Lex2 [16], PopoolTE2 [17]) based on the analyses of: (1) the TEI junction and flanking sequences (split-reads), (2) the paired-end information, (3) the depth of coverage, or (4) a mix of these three criteria. However, these approaches revealed only a portion of the repetitive sequence landscape, and they detect many false positives due to various factors. The first one is linked to the library preparation and PCR amplification that lead to the generation of PCR chimera and thus false positive insertions [18], or to biased sequence representativity (AT- and GC-rich sequences are less represented in Illumina sequencing). The second factor is inherent to the sequencing size (short reads) that does not span more than 400 bp, thus hindering the full sequencing of any repeat or variation larger than this size, especially insertions [19]. The third one is related to the difficulty in detecting TEIs occurring at low frequency in an individual or population. Indeed, these TEIs are usually under-represented in the sequencing data and generally confused with background errors [18]. The comparison of different methods to identify TEIs shows very small levels of overlap [20]. Another weak point of the Illumina sequencing technology is that the insertion size and sequence are not accessible, because this approach generally only gives the global position.

Long-read, or third-generation, sequencing technology might improve the detection of long structural variants and thus of TE variations, and also reduce the detection of false positives/false negatives. This technology should allow for the identification of full copies. Indeed, long-read sequencing methods generate individual reads that are mostly longer (>15 kb) than many of the repeats (TE sequences are generally smaller than 10 kb). Moreover, it solves the problems linked to PCR-based library preparation because it relies on direct DNA sequencing without amplification. However, the main drawback of long-read sequencing, such as the Oxford Nanopore Technology (ONT), is the high rate of single read sequencing errors (3 to 8% for the recent sequencing and base calling) that could introduce bias in data interpretation. This problem is partially solved by increasing the coverage and by improving the final assembly quality by polishing, thus providing an almost perfect genome sequence. Such an approach, based on PacBio sequencing, has already allowed the detection of 38% more TEIs in *Drosophila* chromosome 2 L compared with the available short-read sequencing estimates [21]. Different *Drosophila* genome assemblies using ONT sequencing have also been reported [22,23]. Long-read sequencing methods allow almost complete chromosome-scale genome assemblies, instead of the fragmented draft genomes provided by short reads. Therefore, the assembled individual genomes can be directly compared without the need for any reference genome and their relative structural variants can be scored without biases (or very few). In addition, long-read sequencing of genomes should allow identifying real TEI sites and accurately determining TE copy number at the inter- and intra-population levels. This approach might also help to analyze repetitive regions like PIWI-interacting RNA (piRNA) clusters that contribute to maintaining genome integrity by repressing TE mobility.

Here, we developed strategies to generate *de novo* assemblies of high quality long-read sequencing data, suitable for genomic analyses of TEs present at high and low frequencies in *Drosophila* populations. We first validated our method by comparing the data (genome size, TE content and TEI site estimation) obtained by short and long-read sequencing in *D. melanogaster* and *D. simulans*, two closely related species, but that may vary in TE content [24,25]. We found that, although the *D. simulans* genome contains a large number of old and degraded TE copies, among the most recent pool of insertions, DNA transposons display higher intra-family sequence divergence than LTR elements, suggesting that elements of this group invaded the genome more recently than DNA transposons. Moreover, we observed that piRNA production correlates with TE genome occupancy. When considering the most recent pool of TE insertions, we could not find convincing evidence supporting the piRNA clusters trap model [26,27]. Finally, we developed and validated an approach to identify TEI that occur at low frequencies in a population.

## 2. Materials and Methods

### 2.1. Drosophila Strains

The wild-type *D. melanogaster* and *D. simulans* strains from natural populations were kept at 24 °C in standard laboratory conditions on cornmeal–sugar–yeast–agar medium. The eight samples of *D. melanogaster* and *D. simulans* natural populations were collected using fruit baits in France (Gotheron, 44°56’0”N 04°53’30”E-“goth” lines) and Brazil (Saõ Jose do Rio Preto 20°41’04.3”S 49°21’26.1”W-“sj” lines) in June 2014. Two isofemale strains per species and geographical origin were established directly from gravid females from the field (French *D. melanogaster*: dmgoth63, dmgoth101; Brazilian *D. melanogaster*: dmsj23, dmsj7; French *D. simulans*: dsgoth613, dsgoth31; Brazilian *D. simulans*: dssj27, dssj9). Brothers and sisters were then mated for 30 generations to obtain inbred strains with a very low amount of intra-line genetic variability.

A previously published *D. melanogaster* laboratory line [28] was used for Piwi knockdown (piwi KD) in adult follicle cells. This line carries three components: (i) a GAL4 UAS activator driven by the follicle cell-specific traffic jam (tj)-promoter (tj-GAL4), (ii) an UAS short hairpin(sh)-piwi that induces Piwi RNAi, and (iii) the ubiquitously expressed thermo-sensitive GAL4-inhibitor GAL80^ts^. At 20 °C, GAL80^ts^ sequesters GAL4, preventing sh-piwi expression. At 25 °C, GAL80^ts^ is partially inactive, allowing some GAL4-driven expression of sh-piwi in somatic follicle cells. The resulting partial Piwi depletion allows for the derepression of at least two LTR families (ZAM and gtwin) in follicle cells and their integration as new proviruses in the progeny genome [28]. The polymorphism of this line was partially reduced by isolating a single pair of parents and the line was thereafter stably maintained at 20 °C as a large population (more than 500 progenitors at each generation). The G0 and G0-F100 genomic libraries were prepared shortly after isolation of this line and at the hundredth generation, respectively. Soon after isolation of this isofemale line, a subset of individuals at the pupal to early adult stages was shifted to 25 °C for 5 days, and this was repeated for at least 500 flies for 73 successive generations of partial Piwi KD. Then, after six more generations of stabilization at 20 °C, a third genomic library, called G73, was generated.

### 2.2. Genome Size Estimations

Flow cytometry: genome size was estimated according to [29] using fresh samples of 4-day-old females heads with 10 replicates (five heads per replicate) for each *Drosophila* wild-type strain.

*findGSE:* k-mer distribution was established from the Illumina reads using findGSE [30]. Briefly, adaptors were first removed from the reads with Skewer version 0.2.2 (paired-ends) or NxTrim version 0.4.3-6eb8d5e (mate pairs), when necessary. Reads were then treated essentially as previously described [31] to remove duplicates, filter out reads mapping to reference mitochondrial genomes (GenBank AF200854.1 and AF200828.1 [32]) or microbial contaminants. This allowed for establishing the 21-mer distributions from which genome sizes were estimated using findGSE [30] with default parameters, except for dmsj23 in which the k-mer distribution clearly displayed a peak corresponding to heterozygous regions and was thus treated accordingly.

### 2.3. Illumina Sequencing

Wild-type strains: DNA was extracted from 3 to 5-day-old females for each strain using the Qiagen DNeasy Blood&Tissue kit (# 69506) and following the manufacturer’s instructions. Genomic DNA (1.5 μg) was fragmented for a target insert size of 300 base pairs and sequenced by paired-end Illumina HiSeq (125 bp reads). Library and sequencing were performed by the GeT-PlaGe facility, Génopole Toulouse (France).

### 2.4. DNA Isolation, Oxford Nanopore MinION Sequencing and Base Calling

DNA was extracted from ∼100 males from each wild-type and from thePiwi KD lines using the Qiagen DNeasy Blood&Tissue kit. The genomic DNA quality and quantity were evaluated using a NanoDrop™ One UV-Vis spectrophotometer (Thermo Fisher Scientific, Waltham, MA, USA) and a Qubit^®^ 1.0 Fluorometer (Invitrogen, Carlsbad, CA, USA), respectively. Three micrograms of DNA were repaired using the NEBNext FFPE DNA Repair Mix (NEB M6630). End repair and dA-tailing were performed using the NEBNext End repair/dA-tailing Module (E7546, NEB). Ligation was then performed with the Ligation Sequencing Kit 1D (SQK-LSK108, ONT, for G0, and SQK-LSK109 ONT for wild type strains, G73 and G0-F100 samples). MinION sequencing was performed according to the manufacturer’s guidelines using R9.4.1 flow cells (FLO-MIN106, ONT) and a Nanopore MinIon Mk1b sequencer (ONT) controlled by the ONT MinKNOW software (version 18.3.1 for G0, version 19.05.0.0 for isogenic wild-type strains, and version 19.10.1 for the G73 and G0-F100 samples). Base calling was performed after sequencing using Albacore (version 2.3.3) for G0, and the GPU-enabled guppy basecaller in high accuracy mode for isogenic wild-type strains (version 3.1.5), G73 (version 3.3.3) and G0-F100 samples (version 3.4.4).

### 2.5. TE Content and TEI Site Estimates from Illumina Sequencing

TE abundance was estimated using forward reads and two methods: the TEcount module of TEtools [33] and dnaPipeTE (v1.0.0 and v.1.3.1) [11]. TEcount estimates TE abundance by quantifying reads that map to a set of known TE sequences, here the rosetta fasta file [34]. This tool was run using default parameters and Bowtie2 (v2.2.4) [35,36]. dnaPipeTE assembles repeated sequences from a subsample of reads (<1x) and quantifies reads mapping to these sequences to estimate TE abundance. dnaPipeTE was used with the following parameters: -sample_number 2, -genome_coverage 0.25). Concerning the genome size option, 175 Mb and 147 Mb were used for *D. melanogaster* and *D. simulans* samples, respectively. The rosetta fasta file was used as library [34].

TEIs were detected in Illumina sequencing data using a dedicated mapping-based algorithm similar to that implemented in PoPoolationTE2 [17] with paired-end reads as input, FlyBase reference genomes (ftp://ftp.flybase.net/genomes/Drosophila_melanogaster/dmel_r6.16_FB2017_03/fasta/dmel-all-chromosome-r6.16.fasta.gz and ftp://ftp.flybase.net/genomes/Drosophila_simulans/dsim_r2.02_FB2017_04/gtf/dsim-all-r2.02.gtf.gz), and the TE sequence library at https://github.com/bergmanlab/transposons/raw/e2a12ff708c42dcce5b15d6af290506d78021212/releases/D_mel_transposon_sequence_set_v10.1.fa. Sequencing reads are mapped to the reference genome and TE sequences using Bowtie2 (version 2.3.3) [36]. Then, the algorithm scans the resulting Binary Alignment/Map (BAM) files for pairs in which one end matches to the reference genome, the other end to a TE sequence, and the pair cannot be mapped concordantly to the genome. For each pair, the position of the genome-mappable read is noted, and positions are clustered in order to have no read further apart than 100 bp in that cluster. Each cluster is then interpreted as an insertion, the position of which is the mean of the position of the reads it contains, and the strength of which is evaluated on the basis of the number of reads it contains. For the purpose of this study, only insertions that were supported by at least 50 reads were retained. Unlike PoPoolationTE2, the insertions detected with this procedure correspond to occurrences absent from the reference genome.

### 2.6. Small RNA Extraction and Sequencing

For small RNA sequencing, two replicates per strain were prepared. Small RNA was isolated from 50 pairs of ovaries using HiTrap Q HP anion exchange columns (Cytiva, Velizy-Villacoublay, France) as described in [37], and the eluate was run on a 10% TBE urea gel (Thermo Fisher Scientific). Small RNA size selection (18–50 bp) was performed on gel at the sequencing facility. Quality was checked with the Bioanalyzer small RNA kit (Agilent, Santa Clara, CA, USA). Library construction was performed using the TruSeq Small RNA Library kit (Illumina, San Diego, CA, USA) and sequenced (1 × 50 single reads) on an Illumina HiSeq 4000 at the IGBMC Microarray and Sequencing facility. Adapter sequences were removed using cutadatp [38]. Size selection was then performed using PRINSEQ lite version 0.20.4 [39]. All subsequent analyses were built upon small RNA counts after normalization according to the miRNA amounts, as described in [34].

### 2.7. Genome Assembly

Raw nanopore reads were QC checked using Nanoplot v1.10.2 ( https://github.com/wdecoster/NanoPlot) for sequencing run statistics. Reads with QC < 7 were removed by the sequence provider (Montpellier Genomix, Montpellier, France) before QC. For each dataset, mean length, N50 reads, total reads and bases are listed in Appendix A and Table 1. Reads were submitted to Flye v2.6 [40] with standard options, except --plasmids and -threads 16. Raw contigs were polished using four rounds of RACON v1.3.2 [41] with standard options and 20 threads (-t option; the required mapping was performed using minimap2 [42] v2.16 and -x map-ont -t 20 options). At each step, basic assembly metrics (N50, length, L50) were recorded using Assembly-Stats v1.0.1 (https://github.com/sanger-pathogens/assembly-stats). Once polished, assemblies were visually inspected using D-genies v1.2.0 [43], and incongruencies manually corrected using samtools v1.9.0 [44], faidx command for sequence extraction, and Gepard [45] v1.4.0 for visual determination of breaking points. The corrected assemblies underwent super scaffolding using RaGOO v1.1 [46] with -s (structural variants (SV)) and -t 4, using the specific reference genome (from FlyBase): Dmel_R6.23 for G0 and *D. melanogaster* samples, Dsim_r2.02 for *D. simulans*, and the previously assembled G0 for G73 and G0_F100 samples. Once the assembly was finalized at the chromosome scale, a Benchmarking Universal Single-Copy Orthologs (BUSCO) analysis [47] using the gVolante web service [48] was performed using the BUSCO v2/v3 option and the *Arthropoda* reference set (Figure 1). TE content was estimated in the corresponding chromosome assemblies using RepeatMasker (http://www.repeatmasker.org) and the Dfam database [49].

### 2.8. Global Structural Variant Detection

Global variant detection (i.e., variants common to most genomes of a considered sample compared with the reference genome, see below) was performed using the svTEidentification.py tool (available at https://github.com/DrosophilaGenomeEvolution/TrEMOLO). Briefly, this tool recovers the insertion and deletion positions and creates the associated fasta sequence, based on the Assemblytics report from the RaGOO scaffolding (the deletions are extracted from the reference and the insertions from the new assembly). Once the fasta file corresponding to the SVs was recovered, these sequences were matched with the Basic Local Alignment Search Tool, nucleotide to nucleotide (BLASTN)+ v2.4.0 to a specified TE database. Hits larger than 80% of the TE sequence and identical to more than 80% at the nucleotide level were considered as candidate for new TE insertions/deletions (TEI/TED) in the G0, G0-F100 and G73 samples. For wild-type strains, new insertions/deletions were detected without any filter. The potential candidates were then listed in a tabular format that included their position, size and percentage of size or similarity compared with the reference TEs. The used TE database was a collection of the reference TEs from Bergman’s laboratory (https://github.com/bergmanlab/transposons) and from previously published data [50].

### 2.9. LTR Minor Insertional Variant (LTR MIV) Detection

Each raw long read was mapped using minimap2 v2.16 (-ax map-ont -t 16 as options) to the assembly corresponding to that set of long reads. After recovering the sam file, samtools v1.10.0 was used to compress and sort the sam file in BAM with samtools view and samtools sort (basic options, but with 16 threads), and the MD tag was added using samtools calmd. Then, SV were detected in the resulting sorted BAM file using Sniffles v1.0.10 with at least 1 read and --report_seq -s 1 -n -1 as parameters [51]. These sequences longer than 1000 bp were aligned with BLASTN v2.4.0+ (-outfmt 6) to the LTR subset (60 families) of the database used before. A nucleotide alignment of more than 94% identity and a minimum of 90% of the total length of the TE consensus sequence were then considered as criteria to validate a putative LTR minor insertion variant (LTR MIV), if the length of the variant did not exceed the total size of the TE by more than 18 nt. This corresponds to the largest target site duplication (TSD) ever reported to flank any LTR TE [52]. All codes are available in a snakemake file at https://github.com/DrosophilaGenomeEvolution/TrEMOLO.

### 2.10. Fluorescent In Situ Hybridization on Polytene Chromosomes

Polytene chromosomes were squashed from salivary glands of third instar male larvae. *Not*I and *Pst*I restriction enzymes were used to extract a fragment of the ZAM *pol* gene from a previously published plasmid [53]. The probe was labeled with digoxigenin-11-dUTP using the Nick Translation Mix (Roche #11 745 816 910), and signals were detected with anti-digoxigenin-rhodamine Fab fragments (Roche). The fluorescent in situ hybridization method was adapted from a previously described protocol [54].

### 2.11. Automatic Identification of the Target Site Duplication for LTR MIV

The putative LTR MIVs matching to six LTR families (blood, gtwin, mdg3, ZAM, roo, and copia) were studied. One read supporting each MIV, previously extracted in a fasta file, was compared by BLASTN v2.4.0+ with the corresponding consensus sequence. To automatically check for the presence of a TSD, the positions of the 5′ and 3′ end of the TE alignment were determined within the read. 30nt-long sequences upstream and downstream the putative insertion site were extracted and were aligned to detect the presence, on both sides of the insertion, of a short duplication, the size of which was previously reported by [55] for ZAM and by [52] for the other TEs. The resulting TSD sequences were then extracted and used to create sequence logos with WebLogo (https://github.com/WebLogo/weblogo). All scripts and codes for this automatic extraction are available at the project GitHub.

### 2.12. piRNA Cluster Identification in the Assembled Genomes

To determine the piRNA cluster localization in genome assemblies, a previous annotation of piRNA clusters in the *D. melanogaster* Dmel_R6.04 genome release was used [56]. The flanking genes for each of the 153 major piRNA clusters were identified, their sequence was extracted and mapped to the new reference using BLASTN to locate the limits of the corresponding piRNA clusters in the corresponding assemblies. When only a single gene could be used as border, the piRNA cluster length described in [56] was used to define the other border. Bona fide piRNAs were extracted from the previously published G0 small RNA-seq library [28], and from each of the small RNA-seq libraries presented here, as reads longer than 23 nt that do not map (bowtie --best) to sequences of other known small RNAs (downloaded from FlyBase [57] and MirBase [58]). These selected small RNA reads were then mapped to the corresponding assemblies using Bowtie 1.2.2 [59]. Bowtie parameters were selected to keep only reads that display unique alignments and <2 mismatches (--best -v 2 -m 1). The positions of uniquely mapped reads were determined in the assembly, and sequences with more than 500 reads were conserved and compared to the piRNA cluster coordinates determined in the assembly of that line. Appendix A shows the list of the 42 piRNA clusters corresponding to the best piRNA producers in the G0 line. The coordinates of these 42 regions were then determined in the G73 and G0-F100 assemblies. For wild-type strains, the piRNA abundance was computed within 1 kb windows.

### 2.13. Comparison of ZAM Sequences

After obtaining the corresponding region of the ZAM insertions the fasta sequence was extracted (using bedtools getfasta) and compared with the ZAM sequence at a global level using redotable v1.1.

## 3. Results and Discussion

### 3.1. Using Oxford Nanopore Technologies (ONT) to De Novo Assemble the Highly Contiguous Genomes of Several Isogenic Wild-Type Strains and of one Unstable Line

The ONT-based single-molecule long-read sequencing data provided between 5 and 24 million reads, with a depth of coverage ranging from 40x to 196x (mean = 130x), and a N50 ranging from 3.7 to almost 20 kb (mean = 11 kb) (QC 7 reads only; Appendix A). The N50 large range was explained by the different methods used for genomic DNA extraction and ligation (Materials and Methods). Our assembled genome procedure is summarized in Figure 1. To compare our data with the reference *D. melanogaster* and *D. simulans* genomes, whole genome alignments and local dot plots were performed using D-genies and Gepard, respectively (Appendix A).

A strong correspondence was observed between most de novo assemblies and the corresponding reference genome, except for the G73 and dsgoth31 assemblies in which incongruent contigs were detected. These incongruent contigs were manually broken at the discrepancy points (Appendix A) and the final statistics for the de novo assemblies were obtained using Assembly-Stats (Table 1).

Using our approach based only on ONT data, the N50 ranged from 1.2 Mb (L50 of 33 contigs) to 21 Mb (L50 of 3 contigs). The previously described de novo *D. melanogaster* hybrid assembly obtained using BioNano and assembly merging [23] reported a N50 of 9 Mb (L50 of 6 contigs) for the raw data, and a N50 of 21.3 Mb (L50 of 3 contigs) after merging. Moreover, the BUSCO score of their hybrid assembly was 97.2% after Illumina polishing, while the BUSCO score of our assemblies ranged from 93.7% to almost 99% (98.5% for the reference Dmel_R6.23 assembly [23]) only with RACON polishing. This comparison indicates that our assemblies are of high quality, and that RaGOO use as scaffolder allowed obtaining high-quality assemblies at the chromosome scale.

### 3.2. Estimation of Genome size Using Different Methods

To determine the quality of the ONT-based assemblies of the isogenic wild-type *D. melanogaster* and *D. simulans* genomes, their sizes were compared to the genome sizes estimated with two other approaches: findGSE (based on k-mer estimation) and flow cytometry (Appendix A).

Genome size estimates varied between 142 and 144 Mb (flow cytometry) and 129 and 132 Mb (findGSE) for the *D. simulans* strains and between 162 and 163 Mb (flow cytometry) and 133 and 137 Mb (findGSE) for the *D. melanogaster* strains after excluding dmsj7. The k-mer distribution obtained for this strain was much more scattered than the others, and resulted in a k-mer-based genome size estimate of 147 Mb, most probably an artefact. The size estimated obtained using the ONT data ranged between 131 and 142 Mb for the wild-type *D. simulans* strains and between 130 and 134 Mb for the *D. melanogaster* strains, with similar values for the final assemblies. The correlation coefficients were significant only between the ONT-based and the flow cytometry estimates for *D. melanogaster* (r = 0.9675, *p* = 0.0325), but not *D. simulans* (flow cytometry: r = 0.7564, *p* = 0.2436; findGSE: r = 0.1237, *p* = 0.8763). The correlation only with the flow cytometry estimate indicates that the different genome compositions, and probably the different amounts of heterochromatin affect the estimations obtained by findGSE. The genome size estimates obtained with findGSE were globally more similar than those obtained using the de novo assembly approach, but no correlation was observed between these values, probably due to the different amounts of repeats present in the various strains. In conclusion, genome size estimations present several biases in function of the used method, and ONT assemblies seem to give values close to those obtained by flow cytometry, which is a more global method.

### 3.3. Comparison of TE Abundance in the Isogenic Wild-Type Strains Measured by Illumina and ONT Sequencing

To validate the ONT approach, the TE abundance in the isogenic wild-type *D. melanogaster* and *D. simulans* strains was evaluated using dnaPipeTE [11] and TEcount [33] for Illumina sequencing data, and RepeatMasker for ONT assembled chromosomes (Figure 2). Overall, TE content (expressed as genome percentage) was often higher when estimated using dnaPipeTE (Illumina data) (Wilcoxon matched-pairs signed rank test; *p* = 0.0234) than with RepeatMasker (ONT assemblies) (Wilcoxon matched-pairs signed rank test; *p* = 0.0156). This might be explained by the fact that unlike the RepeatMasker TE database, dnaPipeTE is based on the de novo detection of TEs and the local assembly of TE families, independently of a previously annotated reference genome, thus recovering the maximum number of reads that correspond to known and unknown TEs. In agreement, the correlation was higher between the results obtained with RepeatMasker (ONT data) and the results obtained with TEcount, which is based on the read similarity against a curated database of known TEs [34] (r = 0.8921, *p* < 0.0001), than with dnaPipeTE (r = 0.8504, *p* < 0.0001) (Figure 2, right panel). As previously reported, the LTR group was more abundant than the LINE and DNA transposon groups in all *Drosophila* genomes (see [60] for a review).

### 3.4. Comparison of the TEI Sites Identified in the Isogenic Wild-Type Strains Using the Illumina and ONT Data

Before focusing on the results provided by the ONT approach, we first compared these data to the classically used Illumina results based on discordant pairs of reads (method developed in the laboratory, see Material and Methods). The number of TEI sites tended to be higher when using the Illumina data than ONT data (Wilcoxon paired test, *p*-value = 0.023). This could be due to the presence of false positives caused by PCR artefacts during the Illumina library preparation [18], and/or to the fact that some TEIs might have been too short (fragmented or partially deleted) to be identified using the assembled ONT data. Using the Illumina approach, TEI numbers were significantly lower in the *D. simulans* than in the *D. melanogaster* strains (Wilcoxon test, *p*-value = 0.029), but not when using the ONT data (Wilcoxon test, *p*-value = 0.343) (Figure 3). This may reflect a bias towards *D. melanogaster* sequences in our TE reference file, and/or a long-term difference in TE dynamics between these species [25,61]. Comparisons (chi-square tests) of TEI distributions across TE groups (DNA, LINE, LTR) (see Appendix A) showed that in *D. simulans,* the distributions obtained using both approaches were similar. Conversely, in *D. melanogaster*, the TEI number for retrotransposons was significantly higher relative to the other groups, when using the Illumina approach. This may be due to the higher propensity of *D. melanogaster* retrotransposons to be involved in Illumina PCR chimeras [18] because of their higher genome occupancy (Figure 2), and this difference may be amplified by the exponential behavior of the PCR reaction.

In the subsequent analyses, only TEIs identified using the ONT approach (i.e., the most reliable set of recent insertions) were considered.

### 3.5. TEI Landscape in the Isogenic Wild-Type Strains

Using the ONT approach, the de novo genome assembly of each wild-type strain was compared with the reference genome and the detected insertional structural variants were called global variants (see Figure 1). These global variants correspond to the most recent TEIs. On average, there were 492 and 456 global variants in *D. melanogaster* and *D. simulans,* respectively (Table 2).

DNA transposons were the most abundant group in both species (188 and 215 copies, on average, in *D. melanogaster* and *D. simulans*, respectively), and LTR retrotransposons the least abundant (147 and 117 copies, on average, in *D. melanogaster* and *D. simulans*, respectively). These results may seem in contradiction with the previous data on genome occupancy. However, in this analysis only recent insertions were considered. Moreover, as DNA transposons are in general smaller than LTR retrotransposons, similar levels of genome occupancy correspond to higher copy numbers for DNA transposons than for LTR retrotransposons.

Comparison of the locations of the insertions identified in the chromosome assemblies showed that 22 global variants were present in all four *D. melanogaster* strains, and 23 in all four *D. simulans* strains. These were mainly DNA transposons (*n* = 9 and *n* = 10, respectively). The number of shared pairwise global variants was rather low, roughly 10% of all insertions in most comparisons (Figure 4a). *D. simulans* strains appeared equally distant in terms of insertion sites. Conversely, a geographical structuring could be observed in the *D melanogaster* comparisons: strains from the same population shared more insertion sites than strains from distinct populations.

The mean copy numbers for the different TE families were weakly correlated between *D. melanogaster* and *D. simulans* (Figure 4b) (Spearman rho = 0.33, *p*-value = 1e-4, across 129 TE families). Few families were found in the *D. simulans* strains but not in the *D. melanogaster* strains, and vice versa. In *D.* melanogaster strains, the most abundant families were roo (mean copy number: 24.00), jockey (mean copy number: 48.00), and pogo (mean copy number: 44.25), for LTR retrotransposons, LINEs, and DNA elements, respectively. In *D. simulans*, they were roo (mean copy number: 23.00), Cr1a (mean copy number: 18.50), and hobo (mean copy number: 70.50). In addition, some TE families displayed different copy numbers across strains. For instance, the 297 family had 18 copies in dmgoth63, 6 in dmgoth101, 6 in dmsj23, and 5 in dmsj7. Such patterns are suggestive of recent, independent activations, or even bursts of some families in specific strains, as suggested by in situ hybridization studies in a large number of samples [62]. Kofler et al. (2015) studied TE patterns in *D. melanogaster* and *D. simulans* field samples using Illumina pool-seq data [63]. By computing the insertion frequencies for each family of a subset of 121 TE families, they established that LTR elements were more frequent in *D. melanogaster* than in *D. simulans* populations, whereas DNA transposons were more frequent in *D. simulans* samples. A similar trend was observed in the present work: 147 LTR retrotransposon insertions in *D. melanogaster* and 117 in *D. simulans* (Wilcoxon test *p*-value = 0.343); 188 DNA transposon insertions in *D. melanogaster* and 215 in *D. simulans* (Wilcoxon test *p*-value = 0.029).

### 3.6. Comparison of TE Dynamics in Isogenic Wild-Type D. Melanogaster and D. Simulans Strains by Studying TEI Sequences in ONT Assemblies

The major advantage of the ONT approach is its ability to retrieve whole TEI sequences, while short read-based approaches only give access to TE insertion sites. First, the TEI sizes across strains were compared by parsing the BLAST results at the insertion level and by computing the insertion lengths (Figure 5a). The mean insertion lengths (i.e., fragment sizes) significantly varied among TE groups (2-way ANOVA, *p*-value = 2e-81), but not between species (2-way ANOVA, *p*-value = 0.22). LTR retrotransposons were the largest (mean size = 2692 bp), followed by LINEs (mean size = 1290 bp), and DNA transposons (mean size = 1210 bp). The observed absence of difference between species in these global variants differs from what was previously described. Indeed, for a subset of 15 families, Lerat et al. found that TE copies were more internally deleted (i.e., shorter) in *D. simulans* than in *D. melanogaster* [24]. However, analysis of these 15 families using our ONT data indicated that they displayed, on average, longer fragment sizes compared with the other TE families in *D. melanogaster* (Wilcoxon test, *p*-value = 8e-19), but not in *D. simulans* (Wilcoxon test, *p*-value = 0.34) [24]. This suggests that Lerat et al. 2011 focused on TE families that have particularly large copies in *D. melanogaster* [24], probably because they have been more studied in the past due to their easier analysis by in situ hybridization on polytene chromosomes [7,25,64].

Then, the Refiner module of RepeatModeler (http://www.repeatmasker.org/RepeatModeler) was used to compute the intra-family sequence divergence (average Kimura distance) (Figure 5b). This measure is a proxy of the time passed since the last transposition wave(s). Overall, these distributions were not significantly different between *D. melanogaster* and *D. simulans* and among TE groups (2-way ANOVA; species effect, *p*-value = 0.151; group effect, *p*-value = 0.701), showing that the TE recent dynamics are similar in these two species. However, in *D. simulans,* DNA transposons displayed significantly higher intra-family divergence compared with LTR retrotransposons (Wilcoxon test, *p*-value = 0.023). This suggests that among the most recent transposition events, DNA transposon insertions occurred slightly less recently in *D. simulans*.

Kofler et al. 2015 assumed that population frequencies of TE insertions provide an estimator for the insertion age. However, we find that their population frequencies were not correlated with our measures of intra-family sequence divergence (Spearman correlation coefficients: −0.714 (*p*-value = 0.136) and 0.116 (*p*-value = 0.827) for *D. melanogaster* and *D. simulans*, respectively). We think that intra-family sequence divergence is a more direct estimate of the age of transposition events; however, this discrepancy may also reflect differences in the origins of the sampled flies [61,64]. Alternatively, it may suggest that other factors influence insertion frequencies, besides the age since the initial transposition burst. In addition, our analysis only included TEIs that are not found in the reference genome, i.e., TEIs that result from transposition events more recent than the set-up of the actual populations. Altogether, while the TE ancient dynamics are different between *D. melanogaster* and *D. simulans* [60], the present results suggest that *D. melanogaster* and *D. simulans* TE landscapes are rather similar when comparing only global variants (i.e., the subset of the most recent insertions). As already proposed [25], this may reveal that the colonization of *D. simulans* genome by TEs has now reached a state similar to that of *D. melanogaster,* although it started more recently.

### 3.7. piRNAs, piRNA Clusters and TEIs in Isogenic Wild-Type Strains

Another way to study TE dynamics is to understand the way the production of piRNAs is linked to the TEI type and structure. Indeed, some relationships might exist between piRNA abundance and the recent activity of TEs, estimated by the intra-family sequence divergence. Therefore, piRNA production, TE length and intra-family sequence divergence were analyzed for each TE group and strain. This analysis highlighted a significant TE group effect: piRNA counts were higher in retrotransposon families (LTR elements and LINEs) than in DNA transposon families (*p*-value = 2e-9). Moreover, piRNA counts were significantly and positively correlated with genome occupancy (*p*-value = 5e-7), which strongly depends on TE copy number (Figure 6a). The hypothesis that TE copy numbers determine piRNA abundance was previously suggested in *D. melanogaster* [65,66] and is confirmed here also for *D. simulans*. However, it should be noted that genome occupancy accounts only for 6.2% of the total variation of piRNA counts, indicating that many other factors are involved in TE control.

These observations are also in agreement with the idea that newly integrated copies become piRNA producers [67], and that longer copies produce more piRNAs. It should be noted that retrotransposons are on average longer than DNA transposons.

As ONT assemblies also include piRNA cluster sequences, 42AB and flamenco (the two major piRNA cluster producers in *D. melanogaster*) could be retrieved using their flanking genes (see Material and Methods) [68] from each assembly. Alignment of the uniquely mapped piRNA sequences against the assembly of each wild-type isogenic strain (Figure 6b and Appendix A, black lines) indicated that the regions corresponding to 42AB and flamenco did not display any enrichment in global variant insertion numbers (Figure 6b, gray lines). This indicates that recent TEIs are not specifically enriched in the two major piRNA cluster producers in *D. melanogaster* and *D. simulans* strains. Therefore, the analysis of the de novo assembled genomes to follow the piRNA cluster dynamics in these isogenic wild-type strains did not highlight the previously reported high TEI insertion rate within piRNA clusters [26,50,69,70]. Our data suggests the number of recent TEIs fixed in these piRNA clusters is not different compared with anywhere else in the genome. This discrepancy could be explained by the high frequency of deletions (from several base pairs up to several kilobases) that seems to occur in these regions and that affect ancient TEs, which remain as vestiges in these loci, and also recently inserted TEs [50].

### 3.8. Recent TEIs May Not Be Frequent Enough to Be Incorporated in the Assembled Genomes

To challenge the ONT assembly approach, a bioinformatic analysis was performed to identify recent LTR TEIs that occurred during the last 73 generations (G73) in the unstable Piwi KD line (Materials and Methods and [28]). As a control, to estimate the basal transposition rate when TEs are normally repressed by the functional piRNA pathway, the genome of the hundredth generation (called G0-F100) after establishment of the stable G0 isofemale line was also sequenced. Using the pipeline for detection of global variants (Figure 1), no new ZAM insertion could be detected in the G73 assembled genome compared with the G0 reference genome. This is not consistent with previous data obtained by PCR quantification of the ZAM copy number [28]. Therefore, in situ hybridization analysis was performed to determine whether de novo ZAM insertions were present on polytene chromosomes of G73 male larvae (Figure 7a and Appendix A). This analysis confirmed the presence of the two preexisting ZAM insertions identified on chromosome 2R as global variants in the G0 de novo assembly (compared with the Dmel_R6.23 reference genome). These two insertions were also detected in all three G73 larvae analyzed, as well as many other ZAM signals that were not observed in the G0 samples (Figure 7a and Appendix A). As each of these many G73-specific new ZAM insertions was present in a single larva, they were not incorporated in the G73 de novo assembled genome due to their low frequency, and therefore could not be detected as global variants. Based on the G0 assembled genome, the sequences of the two shared ZAM detected by FISH on chromosome 2R could be accessed. One contained the full length canonical ZAM consensus sequence, while the other displayed an internal deletion (Figure 7b).

### 3.9. A Long Read-Based Pipeline to Detect Low Frequency TEI Polymorphisms

To determine whether ONT can be used to detect TEIs with a frequency not high enough to be recovered in the assembled haplotype, an approach to identify “minor insertional variants” (MIV) was developed (Material and Methods, paragraph 2.9, and Figure 1 (gray)). Minimap2 was used to map each individual long read to the corresponding assembled genome, and Sniffles to obtain the list of variants that had been neglected during the assembling process. Some of the sequences identified as MIVs matched to the 60 canonical LTR TE consensus sequences (Materials and Methods).

As expected, very few LTR MIVs were detected in the G0-F100 “stable line”. Only copia and roo, which have high transposition rates [71], exhibited more than four variants (14 and 22, respectively) among the 51 LTR MIVs detected (Figure 7c). Also in the G73 line, copia and roo were among the more active LTR families (35 and 48 LTR MIVs among the 274 LTR MIVs detected) (Figure 7c). However, two other LTR families, ZAM and gtwin (51 and 93 LTR MIVs, respectively), showed a 50-fold increase in G73 compared with G0-F100, which is more than an order of magnitude higher than what observed for any other LTR family.

The next question was to determine whether the 274 LTR MIVs, present at low frequency in G73, had occurred after the establishment of the isofemale line. Indeed, such insertions could have been already present in G0 at high frequency (and therefore, could have been incorporated in the G0 but not in the G73 assembled genome) or at low frequency (and, therefore, detectable only as MIVs in G0). The first hypothesis was ruled out by comparing global deletions in G73 and G0. Very few G0 insertions were lost in the G73 assembly and they all belonged to five LTR families (mdg3, Transpac, 3S18, blood, and driver) that did not show a large MIV increase in G73 (data not shown). The total absence of LTR MIVs in G0 was not in favor of the second hypothesis.

As a large fraction of the 274 LTR MIVs in G73 were supported by a single read (Figure 7d), the next step was to check whether they were bona fide insertions by looking for insertional hallmarks, such as the target site duplications (TSDs) that occur upon integration as a result of staggered double-strand breaks at this site [72]. Flanking duplications were first detected automatically for each of the top six LTR families (mdg3, blood, copia, roo, ZAM, and gtwin) by aligning the two 30nt-long sequences that flank each putative LTR MIV extracted from the read supporting the variant. This analysis showed that depending on the LTR family, 30–80% of MIVs were flanked by a short duplication of the expected size (4 or 5 nt) (Table 3) [52]. The TSD consensus sequences identified are presented in Figure 7e.

The failure to automatically detect a TSD for the other LTR MIVs could be due to the frequent sequencing errors, a known ONT drawback. When located in the genome-LTR junction region, such errors, which may include several nt-long indels, could impair the automatic detection of the expected TSD, as shown in Appendix A for the manual inspection of the 2R-33863 putative ZAM insertion. Even when junctions are correctly determined, a simple sequencing error in one of the duplicated sequences might prevent their perfect matching. However, it was possible to correct the errors present in these single reads by aligning them with the empty genomic target present on the assembled genome (see, Appendix A). Using this method to manually inspect the sequence of all 51 ZAM variant reads, 48 bona fide insertions were identified, as judged by the presence of the expected 4-nt TSD included in a palindromic GC-rich 6-nt target site motif (TSM) (Figure 7f) [52,55].

Therefore, despite ONT low sequencing accuracy, LTR MIVs could be detected with high sensitivity (insertions present in a population at a frequency <1%, because detected as single reads in a 197x average coverage library) and specificity (FDR of 3/51 = 6%).

### 3.10. Invading LTR Elements Are Not Preferentially Trapped by piRNA Clusters

It is widely assumed that a TE invasion is stopped when a member of the TE family jumps into a piRNA cluster that then triggers the production of piRNAs to repress this TE family (i.e., trap model) [27]. Long-read sequencing data allowed determining whether new insertions accumulated in major piRNA source loci during the 73 generations of LTR TE derepression. Comparison of the 42 major piRNA clusters after their localization in the G0 and G73 assemblies (Appendix A) did not highlight any new TEI into any of these piRNA clusters in the G73 assembled genome. However, new insertions that occurred during the 73 generations of piRNA pathway impairment could still segregate as MIVs in the G73 population. Indeed, among the 274 LTR MIVs present in G73, 6.57% (*n* = 18) were located within the 42 major piRNA producers (Figure 8). However, this proportion was very similar to that of the piRNA cluster size relative to the total de novo assembled genome size (7.36%). Therefore, unlike what expected in the trap model, LTR retrotransposons do not seem to have preferentially accumulated in piRNA clusters during the 73 generations of transposition burst. Specifically, assuming a binomial law with *n* = 274 and *p* = 0.0736 and using a one-tailed test, more than 29 insertions (and not the 18 detected) belonging to many different TE families would have been necessary to validate the hypothesis that piRNA clusters are TE trappers (5% probability threshold).

More than 50% of the LTR MIVs located in piRNA clusters belonged to the gtwin family, suggesting that this family inserts preferentially into piRNA clusters. Indeed, among the 93 gtwin MIVs, 11 (11.8%) were found in piRNA clusters, which is very close to the minimal number (*n* = 12) required to reject the null hypothesis of random insertion in the genome (binomial law with *n* = 93, *p* = 0.0736, and 5% probability threshold). More data on de novo gtwin mobilization are needed to confirm their preferential integration in piRNA clusters during a transposition burst and to support the trap model for this TE family.

## 4. Conclusions

Our work demonstrates that long reads are crucial in order to finely describe TE landscapes at the intra-genome scale. Using isogenic wild-type strains and an unstable line with a succession of transposition bursts, we could characterize the most common TE variants in different strains and identify TE minor variants observed soon after transposition. The parallel analysis of two close species (*D. melanogaster* and *D. simulans*) and two genetic backgrounds allowed us to show that overall, TE recent dynamics are quite similar between species and among strains. However, there is still some strain specificity concerning the identity of the most recently active TE families. ONT is also a powerful tool to investigate the dynamics of piRNA clusters, which are in general inaccessible using short-read sequencing methods. We show here that recent TEIs are not enriched in piRNA clusters, despite recent bursts of TE transposition. Moreover, ONT allows detecting very recent TEIs that are sequenced as singleton reads.

## Figures and Tables

**Figure 1 cells-09-01776-f001:**
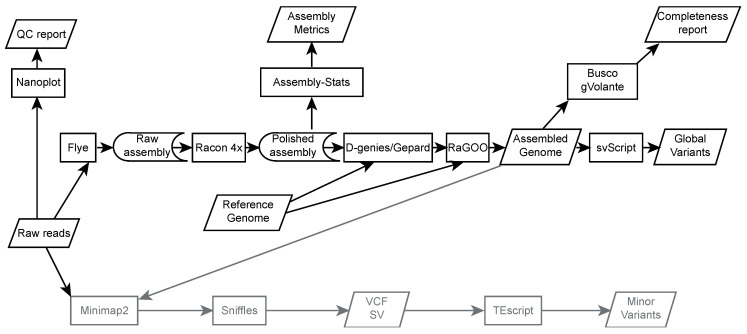
Schematic of the method used for genome assembly and for transposable element insertion (TEI) detection. Global variants (black) were detected from genome assemblies, and minor variants (gray) by remapping reads in these assemblies. The reference genomes used for RaGOO scaffolding were Dmel_R6.23 for G0 and for wild-type *D. melanogaster* strains, Dsim_R2.02 for wild-type *D. simulans* strains, and the G0 assembly for G73 and G0-F100.

**Figure 2 cells-09-01776-f002:**
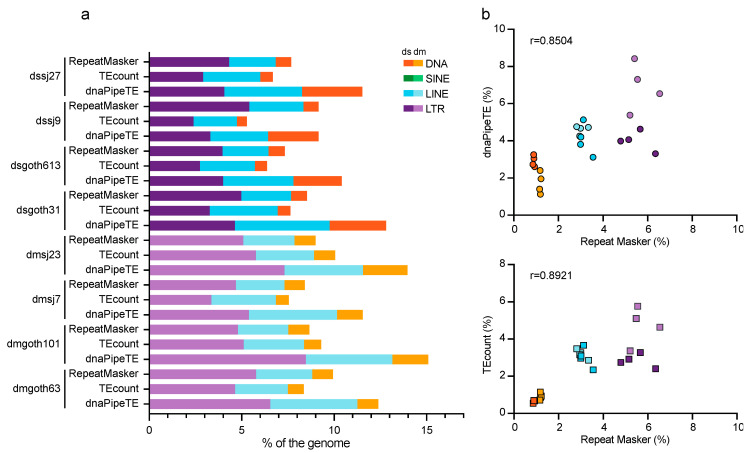
Estimation of the TE percentage in the *D. melanogaster* and *D. simulans* genomes (isogenic wild-type strains). (**a**) Estimation of the TE percentage using RepeatMasker (ONT chromosome assemblies), and dnaPipeTE or TEcount (Illumina reads). (**b**) Correlations between the estimates obtained with the indicated methods.

**Figure 3 cells-09-01776-f003:**
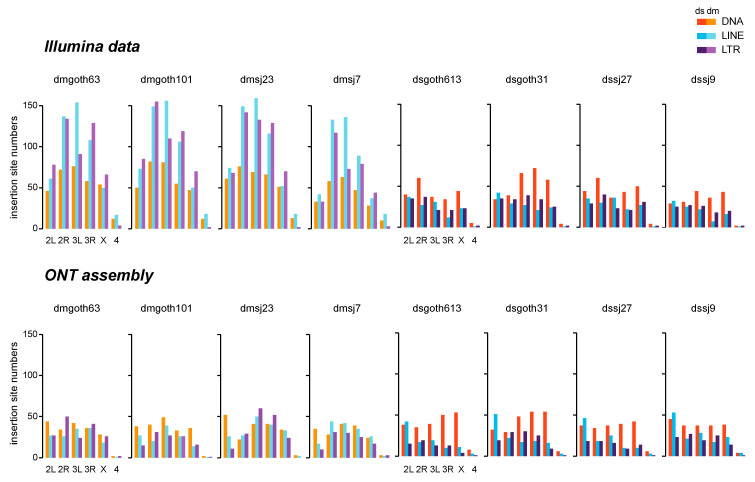
Insertion site numbers for each TE group and per chromosome, determined using Illumina data (upper panels) or Oxford Nanopore Technology (ONT) chromosome assemblies (lower panels).

**Figure 4 cells-09-01776-f004:**
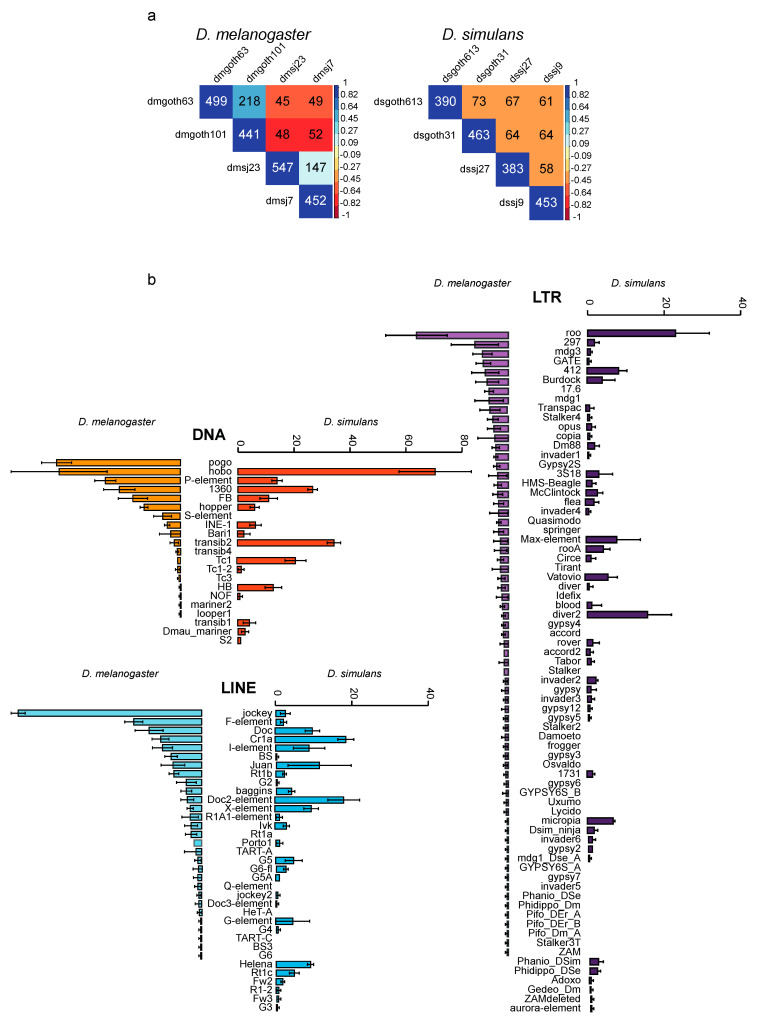
Global variant copy numbers in wild-type *D melanogaster* and *D. simulans* strains. (**a**) Number of shared global variants among strains. The color scale (on the right of each panel) shows the distance based on the number of pairwise shared insertions (indicated in black in the figure). Values in white correspond to the total numbers of the identified insertions for the considered strains. (**b**) Mean TEI numbers for the indicated TE groups computed in the wild-type *D melanogaster* and *D. simulans* strains based on the ONT chromosome assemblies.

**Figure 5 cells-09-01776-f005:**
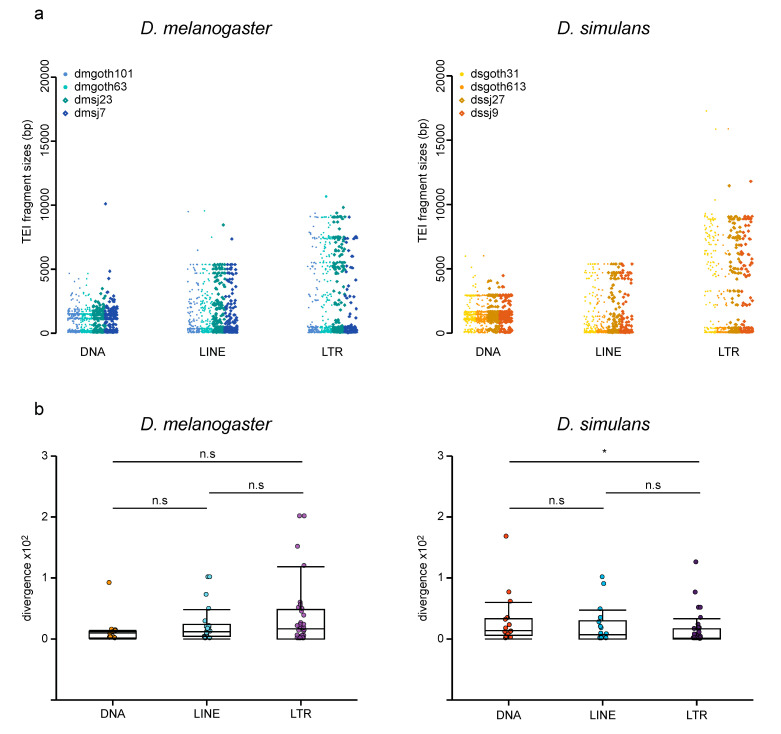
Global variant sequence analysis in wild-type *D. melanogaster* and *D. simulans* strains. (**a**) Distributions of TE copy lengths (i.e., fragment size) in bp for all global variants across strains and TE groups. (**b**) Intra-family sequence divergence (average Kimura distance) computed per strain and per TE family.

**Figure 6 cells-09-01776-f006:**
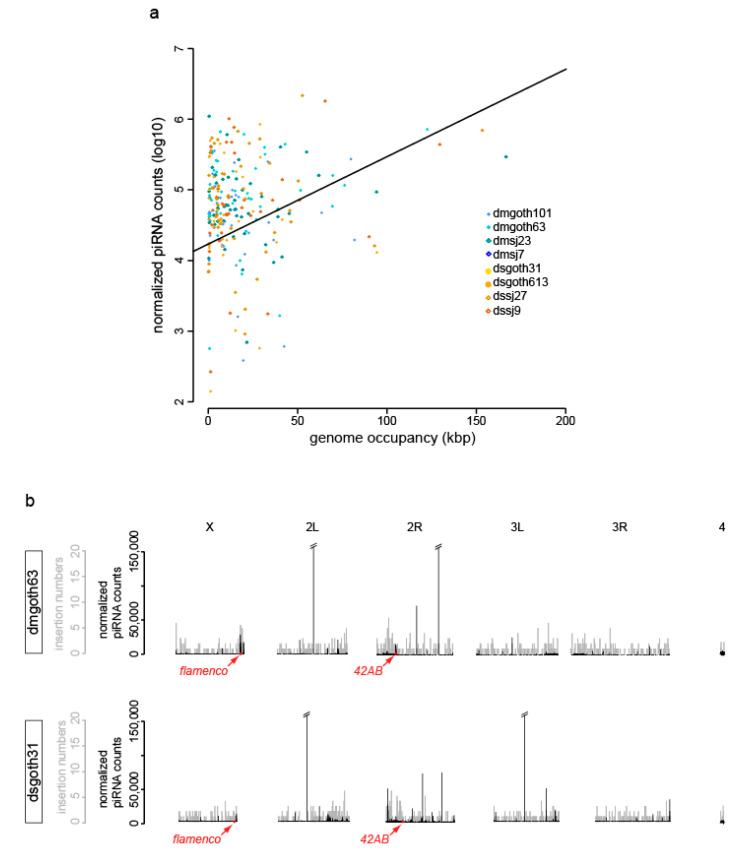
piRNA analyses in wild-type *D. melanogaster* and *D. simulans* strains. (**a**) Normalized piRNA counts (log10) relative to genome occupancy for all strains and the two species and linear regression curve. Each dot is a TE family. (**b**) Results for the dmgoth63 and dsgoth31 strains are shown as examples. Uniquely mapping piRNAs along ONT chromosome assemblies (black, normalized piRNA counts). Global variants identified along ONT chromosome assemblies (gray). Red arrows indicate flamenco (X chromosome) and 42AB (2R chromosome). Data for the other strains are provided in Appendix A. The off-scale peaks might correspond to microRNAs that are absent from miRBase.

**Figure 7 cells-09-01776-f007:**
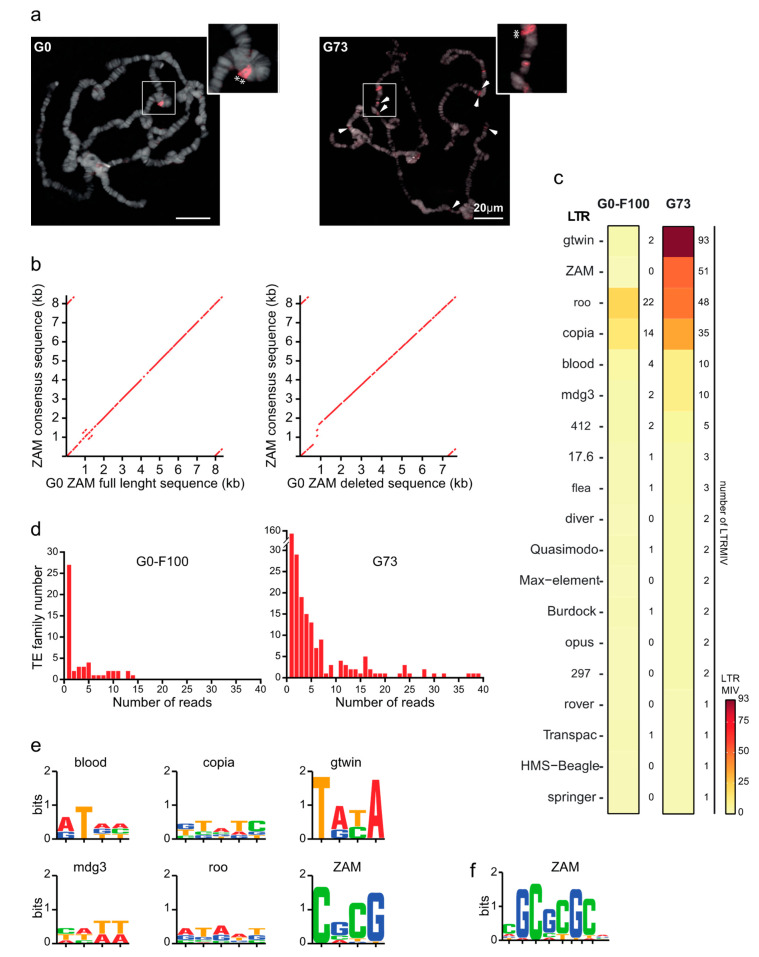
Characterization of the Long-Terminal Repeat minor insertion variant (LTR MIV) in the stable (G0) and unstable (G73) lines. (**a**) ZAM copies visualized by fluorescent in situ hybridization in G0 (left) and G73 (right) polytene chromosomes. The two global variants correspond to non-reference ZAM copies present in G0 and G73 (asterisks in the zoomed images). Arrowheads show the new ZAM insertions in G73. More examples are presented in Appendix A. (**b**) Dot plot of the sequence comparison between the ZAM sequences accessed from the de novo assembled G0 genome and the ZAM consensus sequence. (**c**) Heat map of the LTR MIV detected in the G0-F100 (stable) and G73 (unstable) libraries. (**d**) Histograms showing the number of reads supporting each LTR MIV. (**e**) Sequence logo of TSD defined using the LTR MIV automatic detection procedure. (**f**) the ZAM TSM motif defined using the automatic and manual LTR MIV detection procedures.

**Figure 8 cells-09-01776-f008:**
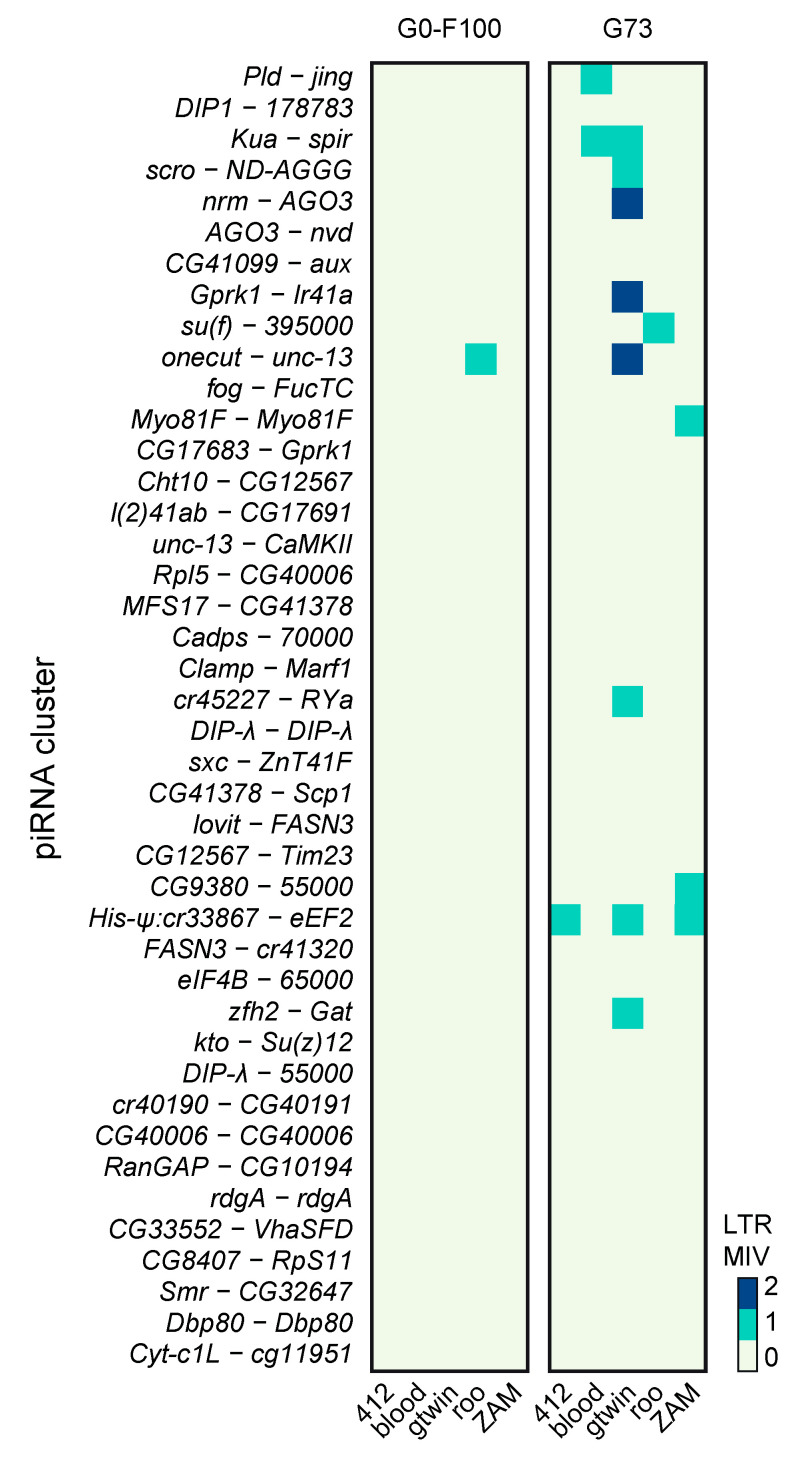
Heat map of the LTR MIVs inserted in piRNA clusters and detected in the G0-F100 and G73 lines.

**Table 1 cells-09-01776-t001:** Statistics for the de novo assemblies before scaffolding. All lengths are expressed in bases. The Benchmarking Universal Single-Copy Orthologs (BUSCO) score indicates the “complete hit” level.

Name	Size	Nb contig	Mean Length	Longest	N50	L50	BUSCO Score, %
dmgoth101	130,483,042	1213	107,571	20,963,225	14,899,963	4	c: 98.6
dmgoth63	134,481,426	1005	133,812	22,615,553	16,996,519	4	c:98.03
dmsj23	131,331,777	1094	120,047	22,945,221	10,553,205	5	c:98.5
dmsj7	131,360,683	1197	109,742	18,094,419	6,212,683	7	c:98.7
dsgoth31	135,039,133	822	164,281	27,577,085	17,530,992	4	c: 98.3
dsgoth613	132,908,190	918	144,78	22,559,698	16,120,890	4	c:98.6
dssj27	134,309,820	866	155,092	27,370,717	20,976,825	3	c:98.6
dssj9	142,009,588	508	279,546	27,589,620	19,611,840	4	c:99
G0	127,415,251	642	198,466	5,037,957	1,208,862	33	c:93.7
G0-F100	139,374,117	836	166,715	17,781,420	9,085,947	6	c:98.97
G73	144,335,962	584	247,15	24,539,270	12,530,957	4	c:98.7

**Table 2 cells-09-01776-t002:** Number of transposable elements insertions (TEIs) identified as global variants in the Oxford Nanopore Technology (ONT) chromosome assemblies.

	dmgoth63	dmgoth101	dmsj23	dmsj7	dsgoth613	dsgoth31	dssj27	dssj9
Total Insertion Number	515	448	550	456	434	496	420	474

**Table 3 cells-09-01776-t003:** Target site duplication (TSD) flanking Long-Terminal Repeat minor insertion variants (LTR MIVs) in the G73 line.

	LTR Family					
	gtwin	roo	ZAM	copia	Blood	mdg3
Total LTR MIV detected (*n*)	93	48	51	35	10	10
TSD automatic detection (*n*)	66	15	25	11	8	5
TSD automatic detection (%)	71	31	49	31	80	50
Additional TSD manually detected (*n*)	NA	NA	23	NA	NA	NA

## Data Availability

Long reads sequencing data used for this study have been deposited at ENA (https://www.ebi.ac.uk/ena) under the accession numbers PRJEB39340 and ERP122844. The small RNA-seq datasets and the Illumina DNA-seq datasets were deposited in NCBI SRA (https://www.ncbi.nlm.nih.gov/sra) under the accession numbers PRJNA644327 and PRJNA644748, respectively. Release 6.23 of the *D. melanogaster* genome and Release 2.2 of the *D. Simulans* used in this study are available on FlyBase (http://www.flybase.org). Bioinformatic scripts and pipelines used for long reads analyses are available at https://github.com/DrosophilaGenomeEvolution/TrEMOLO and for small reads Illumina insertion at https://gitlab.in2p3.fr/pveber/te-insertion-detector/.

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
