# Peer review of "A Transposon Story: From TE Content to TE Dynamic Invasion of Drosophila Genomes Using the Single-Molecule Sequencing Technology from Oxford Nanopore"

_cells, 2020, doi:10.3390/cells9081776_

Round 1

Reviewer 1 Report

The manuscript deals with the identification and analysis of TE and TE insertions at population level using Oxford Nanopore Tech assemblies. Authors provide a very interesting framework where to address these issues and some of their results look interesting.
However, in my opinion, the manuscript needs an extensive revision before to consider it for publication.

General considerations:
Overall, many parts of the manuscript (starting from the Introduction section) seems unnecessarily verbose and require to be shortened. Here and there it is difficult to follow the text but this do not appear related to bad grammar. I have also found particularly difficult to follow paragraphs 3.8 and 3.9, that are actually quite an important part of the analysis. I suggest a deep revision of the text in order to write a more concise and direct text.

A very interesting part of the analyses is the comparison among different approaches to detect TE load. Unfortunately, this is not carried out properly and systematically, and some overstatements about results do not help.
Methods are not described completely and, again, verbosity of the relevant section only make reading vey difficult. The simple use of Repeat Masker without a preliminary de novo search is not the best method to compare with other de novo approaches. It would have been better, for instance, to run an assembly-based de novo detection (like RepeatModeler) and then compare with dnaPipeTE. This, by the way, would be very useful when dealing with high quality draft genomes as it could indicate if read-based approach is similar or may have pitfalls if compared to an assembly-based approach. Moreover, it could give a better indication of the Repeat content in obtained ONT genomes.
Moreover, the lack of statistics in some analyses leaves doubts about some of the statements (see also below points).

TEI: Here the main problem is that it not clear to me why to not consider also insertions present in the reference genome(s). Or, at least, why not analyze them separately in order to have an internal control of the proposed method and a more direct comparison with previously obtained TEI estimates.

Other points:
page 11, lines 243-244: The use a web link as subject of the sentence appears to me incorrect. Better something like "The reference set of TEs was taken at https:..."

page 21, line 477: Probably "home-made" is not the best term to use. you can use term like "species-specific curated TE/repeat collection" or something like that.

page 23, lines 496-497: Authors state that D. simulans show a clear reduction of TEI numbers, but this is not based on any statistics. Looking at figure 3, it seems this is evident only in Illumina-based analyses, while it is less clear in ONT analyses. I suggest to introduce a statistics (probably even a simple Mann-Whitney or a Kruskal-Wallis would do the job) to better discuss this finding.

Page 23, line 504: I may have missed something, but it seems to me that the two approaches (very interestingly) do not reveal similar estimates at all. How do you base this statement?

page 18, lines 396-397: That's just an unnecessary repetition of M&M section. Please, delete.

page 19, lines 432-433: I don't think it is necessary to clarify what N50 is. It is, in fact a well-known, common stats for genomic sequencing.

page 19-20, lines 432-441. I think it is better to state that ONT obtined genome assemblies are of comparable quality to those obtained with BioNano. After all, some stats are better with this latter method and, on average, BioNano looks better.

page 33, lines 687-691: not clear sentence to me. please, rephrase.

Reviewer 2 Report

This manuscript describes an effort to make use of Oxford Nanopore sequencing technology (ONT) to sequence long reads and analyze Drosophila genome. It focuses especially on transposable element (TE) insertions in different species of Drosophila genome, and also piRNA target transposon insertions after depletion of piRNA pathway. The authors start by comparing the TE insertions within genome sequencing data obtained by illumina and ONT sequencing, and concludes that ONT sequencing is more efficient. Moreover, they analyze genomes of different Drosophila species, Drosophila melanogaster and Drosophila simulans, to compare recent TE insertion landscape, reporting that two species were rather similar. Finally, they make use of Drosophila system in order to analyze piRNA clusters and TE insertions, and how they are affected by Piwi-piRNA regulations. The authors describe interesting findings which could be observed due to the use of novel sequencing technology. However, I have several questions and major concerns with the results and the data interpretation.

Major comments:

  1. The first part of the Result and Discussion, “Using ONT to de novo assemble high contiguous genomes of several isogenic wild-type trains and of the unstable line” to “Estimation of genome size of isogenic wild-type strains by different methods”, can be simplified and many of the descriptions can be moved to materials and discussion part. It is quite lengthy, although this part is not the main claim of this paper.

  1. At line 504, authors indicate “TE pools revealed by both approaches are rather similar”, and they decided to use ONT approach. I do not understand the logic here. They say that high number of insertions observed for illumina data analysis was due to artifacts, but can they confirm that? (For example, can some of the illumina specific TE insertions be sanger-sequenced in order to confirm that they actually are artifacts?)

  1. For the comparison between D. melanogaster and D. simulans, could authors rule out the possibility that relatively young TEs (which may vary between individuals) could not be detected due to their variations? (As in case they observed within piRNA pathway depleted flies?) Does that result in underestimation of the difference between these strains?

  1. The authors indicate that “genome occupancy only accounts for 6.2% of the total variation of piRNA counts” (line 653). If they focus not on the whole genome occupancy but occupancy within piRNA clusters, would the percentage increase?

  1. According to Figure 6b, there seems to be regions of piRNA enrichment other than flam and 42AB. Could authors identify novel piRNA clusters using their dataset?

  1. Could authors see the variation within increased insertions between individual cells? As whole, level of TEs such as ZAM and gtwin increased upon piRNA pathway depletion (Figure 7). Does this mean that these types of TEs are always effectively transposed or de-silenced upon piRNA pathway depletion? If so, is there any explanation why these types of TEs are selectively up regulated? Are ZAM and gtwin piRNAs major populations in follicle cells?

  1. Are piRNA populations correlated with increase of insertions in G73? Could the authors compare for example gtwin piRNA populations in G0-F100 and G73?

Minor comments:

  1. The correlation coefficient values indicated in the text can also be indicated in corresponding figures.

  1. For Figures 2, 3, the colors between “ds” and “dm” is quite difficult to distinguish (this may rely on the computer display of individuals), could the authors clarify this?

  1. For Figure 3, it is better to use the same maximum number for y-axis “insertion site numbers”, especially because authors want to compare each other and claim that ONT assembly has less insertion than that of illumina data. It is difficult to compare each other at glance, if they have different y-axis scale.

Round 2

Reviewer 2 Report

Although some of the questions remained unanswered due to technical difficulties, the manuscript has been improved due to the revisions made by the authors.